# Asphaltenes at Oil/Gas Interfaces: Foamability Even with No Significant Surface Activity

**Mélanie Arangalage [1,2,3,4], Jean-Philippe Gingras [2,3,4], Nicolas Passade-Boupat [2,3,4], François Lequeux [1,2,3] and Laurence Talini [1,2,3,\*]** 

[1]    CNRS, Sciences et Ingénierie de la Matière Molle, ESPCI Paris, PSL Research University, Sorbonne Université, 75005 Paris, France; melanie.arangalage@gmail.com (M.A.); francois.lequeux@espci.fr (F.L.)

[2]    Laboratoire Physico-Chimie des Interfaces Complexes, ESPCI, 10 rue Vauquelin, 75005 Paris, France; jean-philippe.gingras@total.com (J.-P.G.); nicolas.passade-boupat@total.com (N.P.-B.)

[3]    Bâtiment CHEMSTARTUP, Route Départementale 817, 64170 Lacq, France

[4]    Total S.A., 64170 Lacq, France

\*    Correspondence: laurence.talini@espci.psl.eu; Tel.: +33-140794679

**Abstract:** In the oil industry, oil foams can be found at different steps from the crude oil treatment to the gas stations. Their lifetime can sometimes reach several hours and be much longer than the residence times available for gas/liquid separation. However, the conditions of formation and stability of such foams have been poorly studied in the literature, in contrast to the foamability of aqueous systems. On the fields, it is currently observed that crude oils enriched with asphaltenes form particularly stable foams. In this work, we have studied the influence of asphaltenes on the foamability of oil mixtures. All the experiments were performed on model systems of crude oils, that-is-to-say decane/toluene mixtures containing asphaltenes at concentrations ranging from 0.01 to 5 wt%. We in particular demonstrate that, within the investigated concentration range, asphaltenes from two different wells do not have any significant surface active properties despite their contribution to the foamability of oil mixtures. We show that the formation of an asphaltene layer at the interface with air that has been evidenced in the past results from solvent evaporation. Using a recently developed experiment based on the Marangoni effect with our model oils, we demonstrate that asphaltenes are not surface active in those oils. We further characterize the oil foamability by measuring the lifetime of the foam formed by blowing nitrogen through the liquid in a column. At concentrations larger than 1 wt%, asphaltenes significantly enhance the foamability of the oil mixtures. Moreover, the closer the asphaltenes are to their limit of precipitation the larger the foamability. However, we evidence that the oil mixtures themselves foam and we show the importance to consider that effect on the foamability. In addition, we observe that the foamability of the asphaltenes solutions unexpectedly varies with the initial height of the liquid in the column. We suggest that, although not significantly modifying the surface tension, the asphaltenes could be trapped at the oil/gas interface and thus prevent bubble coalescence.

**Keywords:** asphaltenes; foamability; layer formation; evaporation; oil foam; bubble lifetime

## 1. Introduction

Crude oils are complex mixtures of hydrocarbons and various organic compounds [1–3] including asphaltenes, which are polyaromatic molecules with side alkane chains. Asphaltenes molecules can self-assemble into clusters of colloidal size (a few tens of nanometers) [4–6]. Usually, asphaltenes are rather defined by their class of solubility than by their chemical formula, which makes them very diverse in chemical nature [6]. By definition, they are species that are insoluble in alkanes

and soluble in aromatic solvents such as benzene or toluene. They therefore can be dispersed in toluene/alkane mixtures, up to a given proportion of alkane, above which they precipitate. Although their physical properties may be as diverse as their chemical nature, asphaltenes are generally known to exhibit remarkable interfacial properties [4]. For example, their behavior at water/oil interfaces has been extensively studied in the literature [7–11], leading to the now commonly accepted idea that asphaltenes are responsible for the stability of water/oil emulsions. Indeed, their high polarity due to the presence of polar groups (carboxylic acids, alcohol, etc.) allows them to adsorb at oil/water interfaces. J. D. McLean et al. [9] demonstrated that this stabilization is increased when approaching the precipitation limit which corresponds to the observation of precipitated asphaltenes with the naked eye when adding some alkane to a mixture of toluene and asphaltenes. Sometimes, it has also been observed that a rigid layer can form [12,13] at the water/oil interface limiting coalescence of oil or water droplets.

Asphaltenes can also adsorb at solid/oil interfaces [4,14–17], especially at the surface of clay, silica or metal oxides particles present in oil wells. For example, the acidic groups of asphaltenes interact with the cations contained in clays. The particles, which wettability is thus modified, can have surface-active properties and increase the stability of crude oil emulsions through a "Pickering" process.

Determining the origin of foamability of oil mixtures is a major stake for the oil industry. On one hand, oil foams may be used as fracturing fluids and their stability is then required and obtained by addition of surfactants [18]; on the other hand unwanted foaming of oil is currently observed and can be a hindrance to extraction and recovery processes. For instance, during the oil/gas separation steps, crude oils initially maintained at high pressure in oil reservoirs, are brought back to atmospheric pressure and vaporization of light elements such as methane or ethane may generate extremely stable foams. The lifetime of these foams can range from several minutes to hours, drastically reducing the yield of the separators, which are not designed for such residence times. Furthermore, oil transport in pipelines can also be affected by foaming phenomena. Actually, under certain flow geometries, foam plugs can form and slow down the transport of crude oil. In the field, it was observed that crude oils from wells containing large amounts of asphaltenes form particularly stable foams. It therefore seems that there is a link between the presence of asphaltenes, foamability and stability of oil/gas interfaces. However, the behavior of asphaltenes at oil/gas interfaces and their influence on the foamability of crude oil remains poorly studied in the literature in contrast to their effects on water/oil or solid/oil interfaces.

F. Bauget et al. [19] studied the effect of asphaltenes on the dynamic and static properties of oil/gas interfaces. They measured the foamability of solutions of asphaltenes in toluene by blowing gas into a glass column containing these solutions. They observed that at high concentrations of asphaltenes (>5 wt%) the foamability increases significantly similarly to the increase in viscosity. Moreover, single film lifetime measurements were performed. They observed that the lifetime of these films increases with the concentration of asphaltenes from 15 wt%. This increase could not be fully explained by the increase of the viscosity. They concluded that, at high concentrations, asphaltenes show surface-active properties at oil/gas interfaces. Surface tension measurements allowed them to support this hypothesis, in particular the significant decrease in surface tension they observed over very long times, of the order of $10^5$ s. In the same pending drop experiments, the formation of a solid shell at the air/drop interface was observed, together with an increase in the surface elasticity. According to the literature [12,19,20], the adsorption kinetics of asphaltenes at liquid/gas interfaces are slow because asphaltenes clusters behave like proteins. After a diffusion and adsorption step, the asphaltenes molecules reorganize into a network structure. It therefore takes a long time to reach a state of equilibrium compared to a purely diffusive process.

Nevertheless, the mechanism of foam stabilization by asphaltenes remains obscure. In particular, in [19], at short times (t < 100 s) no significant decrease in surface tension was reported for asphaltenes concentrations smaller than 5 wt%. That result is in apparent contradiction with the observed foaming

of the same low concentration asphaltene solutions [19] and the fact that the time scales of a foaming process are expected to be within the same range.

In the present work, we focus on relatively dilute (wt% < 5) asphaltenes solutions in pure toluene and a toluene/decane mixture, for which negligible viscosity increases are expected. We report measurements of the surface tension of these solutions using two different techniques. We emphasize how pending drop measurements over long times may be affected by solvent evaporation. We further report foamability measurements of asphaltenes solutions in toluene and decane mixtures, and we evidence that a large part of the foamability results from the foaming of the solvent mixture itself. In addition, we study the influence of the solubility of the asphaltenes on the foamability. Finally, we investigate the effect of the content of asphaltenes on the foamability and we suggest a mechanism explaining why, although not decreasing surface tension, asphaltenes contribute to the stabilization of oil foams.

## 2. Materials and Methods

### 2.1. Materials

We focus on the behavior of asphaltenes extracted from two fields TR and MO. Toluene ($\geq$99.8%), decane ($\geq$99.0%) *n*-heptane ($\geq$99.0%) and pentane ($\geq$99.0%) were supplied from Sigma Aldrich (Saint-Louis, MO, US). All the concentrations are given by weight percentage (wt%). For the extraction of asphaltenes, PTFE filters with a porosity of 5 μm were purchased from Merck (Darmstadt, Germany) For the Marangoni flow experiments hollow glass spheres (HGS) purchased from Dantec Dynamics (Nozay, France), with an average diameter of 10 μm and a density of 1.1, were chosen as tracers and mixed with the liquid bath.

### 2.2. Methods

#### 2.2.1. Asphaltenes Extraction

The asphaltenes were extracted from crude oils by introducing an excess of n-heptane at room temperature with proportions by weight 1:20. After 48 h of stirring, asphaltenes were collected by vacuum filtration and dried under a hood for 72 h.

#### 2.2.2. Crude Oil Model Solutions Preparation

All the studied solutions were made by dispersing the extracted asphaltenes in decane/toluene mixtures under vigorous stirring during 72 h. We note that TR asphaltenes precipitate in decane/toluene mixtures from 50 wt% in decane and MO asphaltenes precipitate from 40 wt% in decane. TR asphaltenes correspond to the "precipitated asphaltenes from crude oil A" which polydispersity in solubility is described in detail elsewhere [21]. The precipitation rates were determined using the Asphaltenes Solubility Class Index method (ASCI) previously developed [21]. This methodology is based on the precipitation threshold of asphaltenes in different alkane/toluene mixtures which alkane weight percentage varies from 0 to 100 wt% by 5 wt% steps. The limit of precipitation of asphaltenes was estimated with the naked eye.

#### 2.2.3. Tensiometry for Surface Tension and Evaporation Measurements

Surface tension measurements were performed using a tensiometer (Tracker, Teclis Scientific, Tassin la Demi Lune, France). The solutions being opaque, the pendant drop method was chosen. Drops of 11 μL volume were formed with a syringe and a needle of inner diameter 1.19 mm. The apparatus analyses the shape of the drop and the surface tension is obtained after resolution of the Laplace's equation. A motor controls the volume of the drop generated by the tensiometer by image analysis and compensate any volume modification by injecting the solution. Thus, this apparatus provides an easy way to measure the volume of solvent evaporating as a function of time. All the

experiments were performed at room temperature and atmospheric pressure. Two solutions were used: pure toluene containing 5 wt% of asphaltenes TR and a decane/toluene (45/55 wt%) mixture containing 5 wt% of asphaltenes TR.

### 2.2.4. Marangoni Flow Method

The highly sensitive method described in reference [22] was used in order to detect the surface activity of asphaltenes at oil/air interfaces. The original experimental setup consists in the generation of a Marangoni flow at a liquid/air interface by suspending a droplet of solute over a liquid bath. A concentration gradient is generated by condensation of the evaporated solute and, with the surface tension of the two liquids being different, a surface tension gradient forms at the interface. The generated flow is modified in the presence of surface active agents. The addition of a very small amount of surfactants can turn the flow into an oscillatory flow allowing a sensitive detection of traces of surfactants. First we reproduced the experiment described in [22] on solutions of tetradecyltrimethylammonium bromide (TTAB—Sigma Aldrich) in deionized water (cmc of TTAB is $5.0 \times 10^{-3}$ mol/L). The setup consisted of a liquid bath of cylindrical shape containing tracers above which a droplet of ethanol was formed with a neMESYS high precision syringe pump (Cetoni, Korbussen, Germany). We quantitatively measured the flow rate $\alpha$ in the region close to the symmetric axis according to the method described in [22] using a high speed camera (Mini AX, Photron, Tokyo, Japan) and the plug-in ParticleTracker for ImageJ [23]. For the oil mixtures containing asphaltenes studied in this work, we adapted the setup by using a droplet of pentane. Indeed, pentane is a volatile solvent which is soluble in toluene or decane/toluene mixtures. In the case of highly opaque solutions, a 2D-version of the setup was used. The cylindrical liquid bath is replaced by a rectangular tank (section 6.9 × 2.2 cm and thickness 2 mm) consisting of two glass slides clipped on both sides of a Teflon frame. The motion of the tracers are visualized using a laser sheet (StingRay 640 nm, Coherent Inc., Santa Clara, Ca, US) placed vertically as described in Figure 1. The experiments were performed on mixtures of decane/toluene containing different weight percentage of decane and different concentration of either asphaltenes TR or MO at their limit of precipitation:

- A decane/toluene (45/55 wt%) mixture containing 0.0025, 0.05, 0.1, 1.5 or 5 wt% of asphaltenes TR
- toluene containing 2 or 5 wt% of asphaltenes TR
- A decane/toluene (35/65 wt%) mixture containing 0.02, 0.3, 0.1 or 1.5 wt% of asphaltenes MO

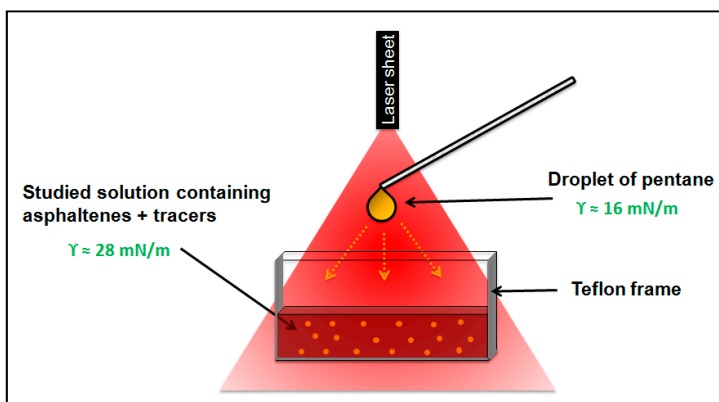

**Figure 1.** Scheme of the Marangoni setup. A droplet of pentane is formed above a liquid bath. The liquid is seeded with tracers (not to scale) which are illuminated by a vertical laser sheet.

### 2.2.5. Foamability Tests

Classical foaming tests were performed in order to measure the foamability of the different solutions. The experimental setup consisted of a glass column (Glass column: 85 mL, ∞ diameter:

2.1 cm, Robu, Hattert, Germany), which lower part is formed of a fritted glass filter (porosity: 10–16 μm). Once the studied mixture was placed inside the column, nitrogen was injected through the fritted glass. The flow rate was controlled using a flowmeter (Fisher Scientific, Hampton, NH, US). A glass reservoir was placed between the column and the flowmeter to prevent backflows of liquid. As the experiments were performed under a hood, evaporation was prevented by placing a glass slide on the top of the open column. Images of the experiment are captured using a XCD-U100 camera (Sony, Toky, Japan) at 30 fps. We have checked that similar foamability results were obtained by injecting nitrogen saturated by the solvent. The initial height $H_0$ of liquid and the final height $H_f$ of foam are listed, the final height being reached a dozen of seconds after the beginning of the experiment.

During the injection of nitrogen, a stationary state was reached within a few seconds and the column was then divided into two parts (Figure 2):

- The lower part consisted of a bubbly liquid of height $H_l$.
- The upper part consisted of a foam of height $H_m$.

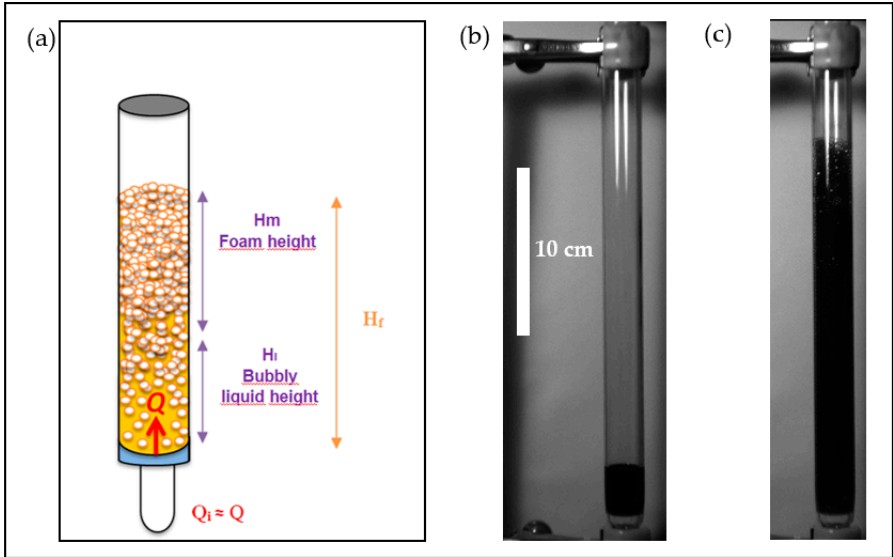

**Figure 2.** Description of the foamability test. (**a**) Scheme of the liquid column during a foamability test. The upper part of the column consists of a foam of height $H_m$ and the lower part of the column consists of a bubbly liquid of height $H_l$. (**b**,**c**) are photographs taken before (**b**) and during (**c**) a foamability test of a mixture of decane/toluene (45/55 wt%) containing 5 wt% of asphaltenes. The foamability is measured at a nitrogen flow rate of $5 \times 10^{-6}$ m$^3$/s and at an initial height of liquid introduced $H_0$ equal to 2.8 cm.

Since it was difficult to observe the boundary between the bubbly liquid and the foam, we rather measured the final height $H_f$ and extrapolated the foam height $H_m$ from the final height $H_f$, using a treatment described in the Appendix A.

In addition, rather than a foam height $H_m$, the foamability can be evaluated as a lifetime $\tau_v$ :

$$\tau_v = \frac{H_m}{U} = \frac{\pi R^2}{Q} \left( \frac{H_f(1 - A) - H_0}{1 - A - \varphi_l} \right) \tag{1}$$

With $U$ the speed of the flow in the column, $R$ the radius of the glass column, $Q$ the flow rate of nitrogen injection, A the volume fraction of gas in the bubbly liquid and $\varphi_l$ the volume fraction of liquid in the foam. The calculation of the lifetime $\tau_v$ (resp. the volume fraction of liquid in the foam $\varphi_l$) is described in the Appendix B (resp. Appendix C). The lifetime $\tau_v$ is an intrinsic quantity which should not depend on the foaming conditions, i.e., neither on the initial liquid height $H_0$ nor on the flow rate $Q$.

## 2.3. Summary of the Investigated Systems and Used Techniques

For the sake of clarity, we summarize in Table 1. the sytems that were investigated together with the different measurements that were performed.

**Table 1.** Summary of the different investigated systems and experimental conditions.

| Experiment | Physical Value Measured | Decane/Toluene Ratio | Concentration of Asphaltenes |
|---|---|---|---|
| Tensiometer (pendant drop method) | Surface tension and evaporation measurements | Toluene<br>Decane/Toluene(45/55 wt%) | 5 wt% TR<br>5 wt% TR |
| Marangoni experiment | Surface activity at liquid/gas interface | Decane/Toluene (45/55 wt%) | 0.0025, 0.05, 0.1, 1 and 5 wt% TR |
| Foamability test | Impact of asphaltenes concentration on foamability | Decane/Toluene (45/55 wt%)<br>Decane/Toluene (35/65 wt%) | Between 0.001 and 5 wt% TR<br>Between 0.01 and 5 wt% MO |
| Foamability test | Impact of asphaltenes solubility on foamability | Decane/Toluene varies from 0 to 100 wt% in decane | 5 wt% TR<br>3 wt% MO |
| Foamability test | Impact of the initial liquid height introduced on foamability | Decane/Toluene (45/55 wt%) | 5 wt% TR |

## 3. Results

### 3.1. Surface Tension for Oil Mixtures Containing Asphaltenes: Layer Formation and Evaporation

We have performed surface tension measurements with a pending drop apparatus that are reported in Figure 3. No significant variation of surface tension was measured over short times (<100 s) whatever the solvent composition and asphaltenes concentration. That result is in agreement with reports of surface tension measurements [19] made with asphaltenes of an origin differing from the ones of the present work.

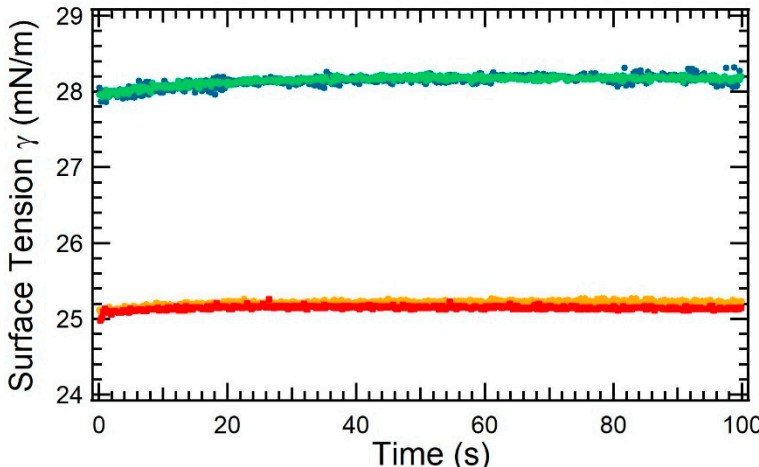

**Figure 3.** Top curve: Surface tension measurements obtained with the pending drop tensiometer of a solution of toluene without asphaltenes (green/light symbols) and with 5 wt% of asphaltenes TR (blue/dark symbols). Bottom curve: same for a mixture of decane/toluene without asphaltenes (yellow/light symbols) and with 5 wt% of aphaltenes TR (red/dark symbols).

Over longer time scales, the surface tension was observed to vary. That effect was systematically associated with the formation of a solid layer at the surface of the droplet, which prevents proper surface tension measurements. The layer becomes solid like with elapsed time and it can be observed by gradually deflating the droplet formed by the tensiometer as shown in Figure 4. A photograph of the remaining layer after deposition of the drop on an absorbing paper is shown in Figure 5.

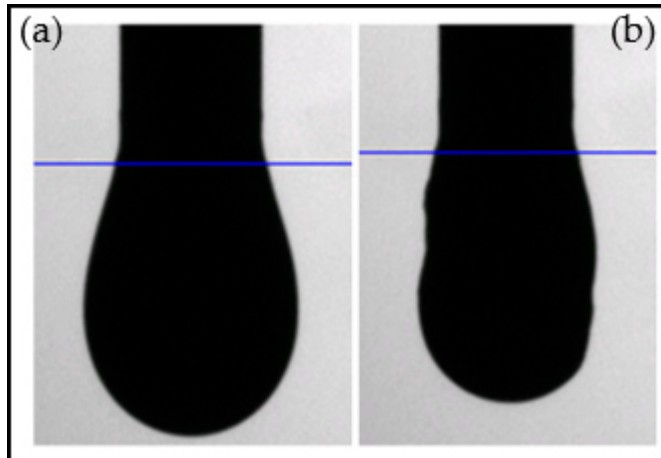

**Figure 4.** Photographs of a droplet of decane/toluene (45/55 wt%) mixture containing 5 wt% of asphaltenes TR at t = 2500 s. (**a**) before and (**b**) after deflation of the drop: observation of an asphaltene layer at the surface of the drop.

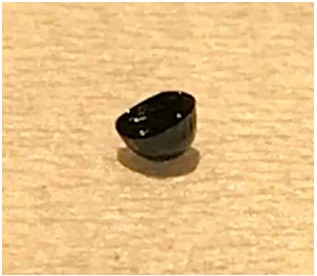

**Figure 5.** Photograph of an asphaltenes layer formed at the surface of the drop of decane/toluene (45/55 wt%) containing 5 wt% of asphaltenes at t = 2500 s.

We attribute the layer formation to an effect of evaporation. Actually, toluene and decane kept evaporating during the experiment, and to maintain the drop volume constant, solvent was continuously injected, as schematically described in Figure 6.

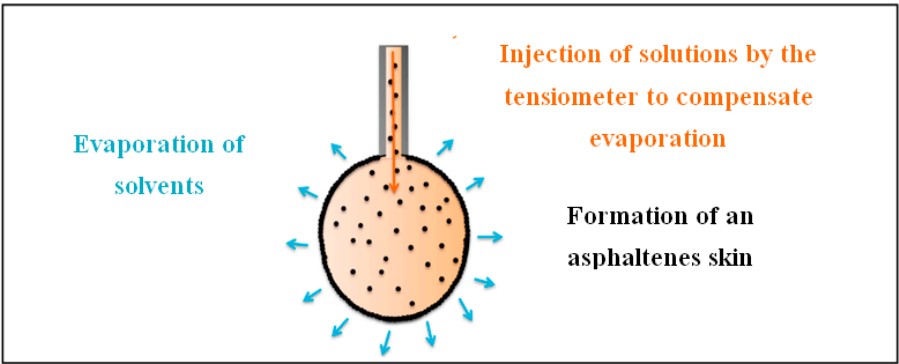

**Figure 6.** Scheme explaining the evaporation of solvents on the surface of the droplet, the injection of the mixture by the tensiometer to compensate this effect and the formation of a layer of asphaltenes.

Over time, the convective flow due to solvent evaporation tends to accumulate the asphaltenes at the surface of the droplet, while diffusion disperses them. The competition between these two effects is described by the Peclet number, which, over a length scale given by the drop radius $R$, is given by:

$$P_e = \frac{\left(\frac{R^2}{D}\right)}{\frac{4/3\pi R^3}{Q_{ev}}} \approx 150 \gg 1 \tag{2}$$

with $D = kT/(6\pi\eta r)$ the diffusion coefficient of a spherical particle of radius $r$ in the liquid of viscosity $\eta$, $k$ the Boltzmann constant, $T$ the temperature and $Q_{ev}$ the evaporation rate measured using the tensiometer ($Q_{ev}$= 30 nL/s for a mixture of decane/toluene (45/55 wt%) containing 5 wt% of asphaltenes TR).

The large value of the Peclet number compared to unity shows that evaporation dominates diffusion. Thus, diffusion is not efficient enough to rehomogenize the concentration of asphaltenes inside the drop. As a result, solvent evaporation accumulates the asphaltenes at the interface and a layer is further formed.

To confirm that picture, we have performed the same experiment within limited evaporation conditions: the drop was placed in a toluene-saturated atmosphere to limit the evaporation of the solvent. That experiment was carried out with a mixture of decane/toluene (45/55 wt%) containing 5 wt% asphaltenes. No layer formation was observed over times longer by 1.5 times than the time at which the layer appeared in experiments with significant evaporation. Owing to condensation droplets appearing on the visualization window of the tensiometer, no information on the behavior over longer timescales or quantitative data could be obtained from that experiment. However, it shows that layer formation results from the evaporation of toluene from the droplet.

In conclusion, in agreement with previously reported data [19], no significant surface tension variation could be measured with classical tensiometry at short times, i.e., at time scales similar to lifetimes of the foams studied herein (<100 s) [19]. In addition, we demonstrate that layer formation at the surface of the droplet does not result from a slow adsorption process of asphaltenes but can be fully accounted for by evaporation of the solvent that induces an accumulation of the asphaltenes at that surface. In the following, we investigate the surface properties of solutions of asphaltenes, using a method which can detect surface tension variations as small as a few $10^{-3}$ mN/m, i.e., 100 times smaller than the limit of classical tensiometry.

### 3.2. Evaluation of the Surface Activity with the Marangoni Flow Method

We now present the results obtained with a Marangoni-flow-based technique in order to determine if the asphaltenes present surface active properties at oil/air interfaces. As detailed above, the quantity that is measured in that experiment is a flow rate $\alpha$ which value depends on the concentration of surface active components since enough surface active components can stop the continuous Marangoni flow and make the value of $\alpha$ zero.

We first show the results obtained with aqueous solutions of soluble surfactants, namely TTAB. Figure 7 is a comparison between the value of the flow rate $\alpha$ in the Marangoni flow method and the foamability of aqueous solutions of TTAB as a function of the concentration of added surfactant. The foamability will be defined later in the paper, here it is considered to be a simple indicator of foam formation. We notice that from $5 \times 10^{-4}$ cmc the Marangoni flow induced by the setup slows down until reaching the oscillating area from $5 \times 10^{-3}$ cmc where by convention the flow rate $\alpha$ is equal to zero. The foamability of aqueous solutions of TTAB is observed for concentrations above $5 \times 10^{-3}$ cmc showing that it is correlated with the threshold concentration of detection of the surface activity of TTAB. We show in what follows that no such correlation can be observed with asphaltene solutions.

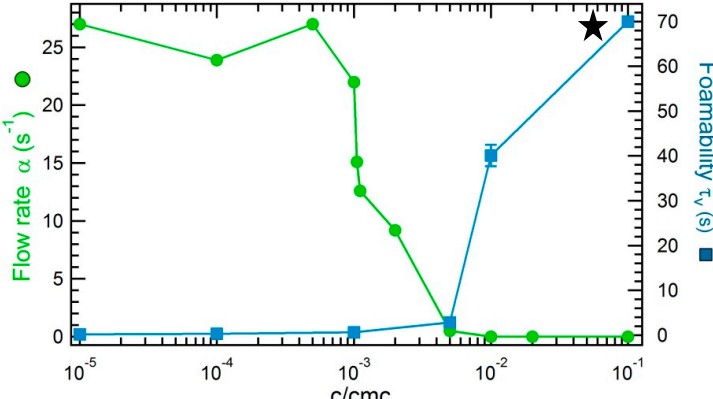

**Figure 7.** Velocity rate $\alpha$ obtained with the Marangoni flow setup and foamability $\tau_v$ as a function of the concentration of the added surfactant for solutions of TTAB in water. The velocity rate was measured at the surface of a bath containing the studied solutions above which a droplet of ethanol is suspended. The zero values of $\alpha$ correspond to the suppression of steady flow. The value of the foamability obtained for the solution at $10^{-1}$ cmc is only indicative since the foam generated is very stable and overfills the glass column: the arbitrary value of the foamability is indicated by a black star.

The comparison between the flow rate $\alpha$ in the Marangoni flow method and the foamability of decane/toluene (45/55 wt%) mixtures as a function of the concentration of asphaltenes dispersed is represented in Figure 8. The solutions containing asphaltenes being opaque quantitative measurements cannot be performed. We arbitrarily assign the qualitative value 1 to the rate $\alpha$ if a continuous Marangoni flow is induced and the value 0 if the flow is oscillatory or stopped. As shown in Figure 8, the Marangoni flow is always continuous whatever the concentration of asphaltenes, their state of aggregation or their origin (MO or TR). This confirms that the asphaltenes investigated herein do not have any surface active property at the oil/air interface in contrast to the commonly accepted idea in the oil industry. However, as shown in Figure 8, asphaltenes solutions do foam significantly. We show in the following that the contribution to foaming of the asphaltenes is significant only for solutions of concentrations larger than 1 wt%.

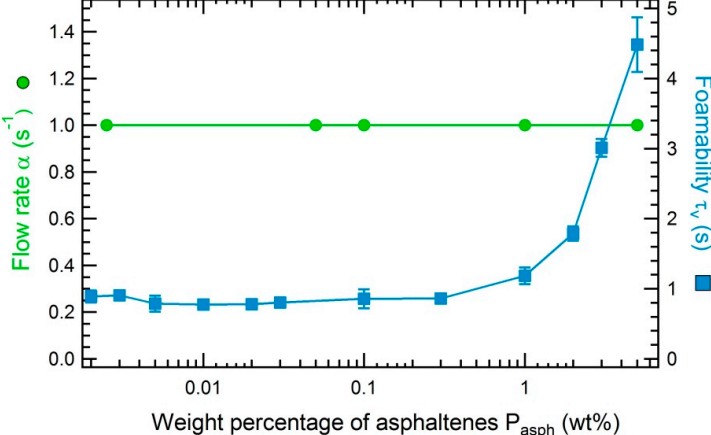

**Figure 8.** Velocity rate $\alpha$ obtained with the Marangoni flow setup and foamability $\tau_v$ as a function of the concentration of asphaltenes TR for solution of asphaltenes in decane/toluene mixtures (45/55 wt%). The velocity rate is measured at the surface of a bath containing the studied solutions above which a droplet of pentane is suspended. The values of the flow rate is only qualitative. We assign the value 1 if the Marangoni flow is continous and the value 0 if the flow is stopped. The foamability tests were performed at a flow rate of nitrogen equal to $1.04 \times 10^{-5}$ m$^3$/s and an initial height of liquid of 1.8 cm.

*3.3. Foamability*

3.3.1. Impact of the Concentration of Asphaltenes on the Foamability

To study the influence of asphaltene concentration on the foaming of asphaltene solution, we have measured the foamability of different solutions and expressed it as a lifetime $\tau_v$. As previously mentioned, that quantity is expected to be intrinsic and to be independent of the foaming experiment conditions, namely the initial liquid height and the gas flow rate. As shown in Figure 9, the foaming behavior is similar for both natures of asphaltenes: we observe that the foamability has a non zero constant value at small asphaltenes concentrations and significantly increases for concentrations larger than 1 wt%. In the investigated concentration range, no significant change of the viscosity by the asphaltenes is expected [19]. Both asphaltenes are placed in a solvent such as to be at their precipitation limit, which corresponds to different proportions of the decane/toluene mixtures for the different asphaltenes. Therefore, we plot the ratio between the foamability of the mixture $\tau_{v\ mix}$ and the foamability of the mixture containing no asphaltenes $\tau_{ref}$. Using that representation, the foamability not only displays the same behavior but also similar values regardless of the origin of the considered asphaltenes.

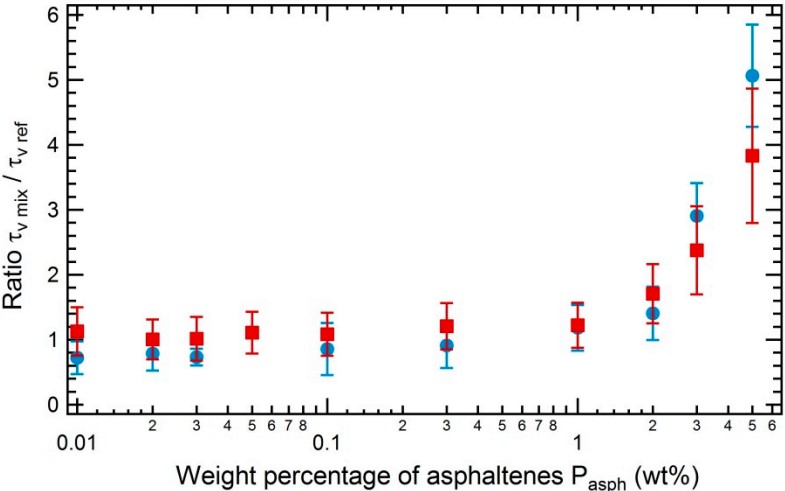

**Figure 9.** Ratio of foamability between the foamability of the mixture containing asphaltenes $\tau_{v\ mix}$ and the foamability of the mixture containing no asphaltenes $\tau_{ref}$ as a function of the weight percentage of asphaltenes dispersed in the mixture $P_{asph}$: asphaltenes TR dispersed in decane/toluene (45/55 wt%) (blue circles) and asphaltenes MO dispersed in decane/toluene (35/65 wt%) (red squares). The flow rate of nitrogen is equal to $5 \times 10^{-6}$ m$^3$/s and the initial height of liquid introduced $H_0$ is 1.8 cm.

In the following section, we study the influence of the composition of the decane/toluene mixture on the foamability of asphaltene solutions. We in particular show that the mixtures without asphaltenes also have a non zero foamability, which corresponds to the times $\tau_{ref}$ used above.

3.3.2. Impact of the Solubility of Asphaltenes on the Foamability

We investigate the effect of the solubility of asphaltenes on the foamability of decane/toluene mixtures by varying the weight percentage of decane at fixed concentration of asphaltenes TR or MO (Figures 10 and 11). For both asphaltenes, a maximum of foamability is observed when approaching the precipitation limit. This confirms the results of N. N. Zaki et al. [24] who observed in similar experiments an increase of foamability as the asphaltenes precipitation limit was closer.

In addition, we report in both figures the foamability data obtained with the asphaltene-free oil mixtures (red circles). The foamability of the oil mixtures is smaller than the one of asphaltene solutions but not zero as would be expected for a pure liquid. The foaming of liquid mixtures has

been evidenced decades ago [25] although it was not fully understood then; it will be the object of a forthcoming paper. To our knowledge, that effect had never been evidenced in the literature on asphaltenes, and we emphasize that, since it is of the same order of magnitude as the foamability of asphaltene solutions, it should be taken into account in order to properly quantify the contribution to foamability of asphaltenes.

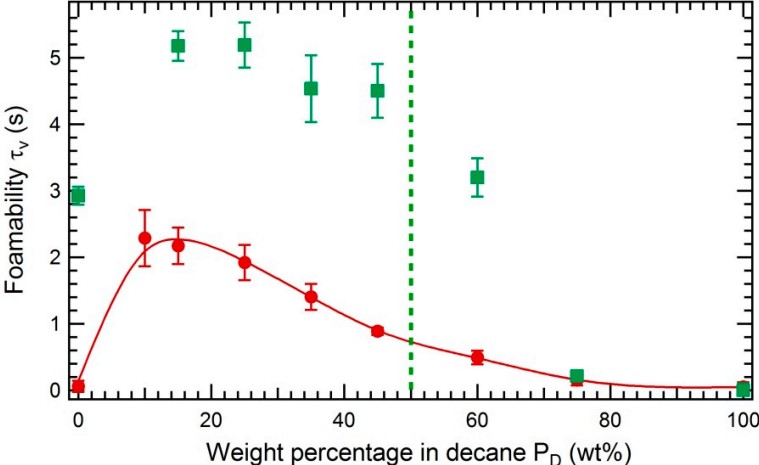

**Figure 10.** Foamability $\tau_v$ as a function of the weight percentage of decane $P_D$: mixtures of decane/toluene (red circles) and mixtures of decane/toluene with 5 wt% of asphaltenes TR (green squares). The green dotted line corresponds to the precipitation limit of asphaltenes TR in decane/toluene mixture. The precipitation limit is 50 wt% of decane. The foamability tests were performed at a flow rate of nitrogen equal to $1.04 \times 10^{-5}$ m$^3$/s and an initial height of liquid of 1.8 cm.

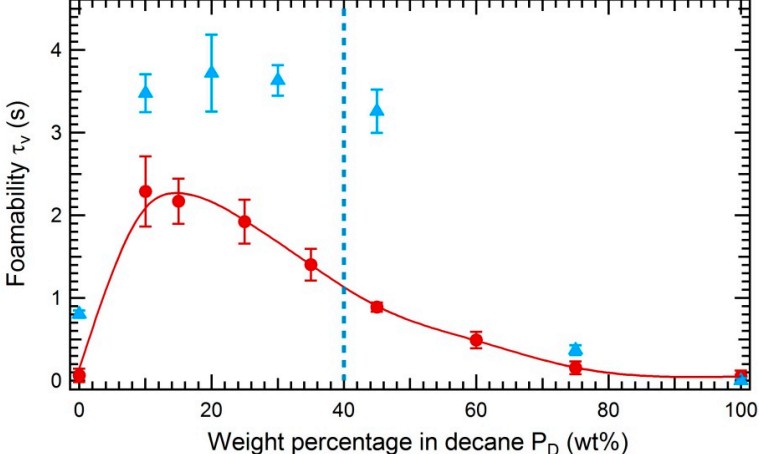

**Figure 11.** Foamability $\tau_v$ as a function of the weight percentage of decane $P_D$: mixtures of decane/toluene (red circles) and mixtures of decane/toluene with 3 wt% of asphaltenes MO (blue triangles). The blue dotted line corresponds to the precipitation limit of asphaltenes MO in decane/toluene mixture. The precipitation limit is 40 wt% of decane. The foamability tests were performed at a flow rate of nitrogen equal to $1.04 \times 10^{-5}$ m$^3$/s and an initial height of liquid of 1.8 cm.

In summary, the foamability of asphaltenes solutions is maximal close to the precipitation limit of asphaltenes. To properly quantify that effect, the contribution to foamability of the oil mixture without asphaltene should be taken into account.

### 3.3.3. Impact of the Initial Height $H_0$ on the Foamability

We have measured the foamability expressed in lifetime $\tau_v$ as a function of the initial height $H_0$ of the liquid introduced in the glass column. The experimental results are shown in Figure 12. In the case of an asphaltene-free decane/toluene mixture, the foamability does not vary with the initial height $H_0$, as expected from the definition of $\tau_v$ which is directly related to the lifetime of a bubble in the foam. Surprisingly, a significant increase of $\tau_v$ with initial height was observed with the asphaltene solution. Since it means that the foamability depends on the quantity of asphaltenes present in the column, evaporation can be suspected to be at the origin of that effect. However, experiments performed with nitrogen saturated with toluene vapor showed no significantly different foamability. We can therefore be confident that evaporation of toluene is not at the origin of the behavior observed in Figure 12.

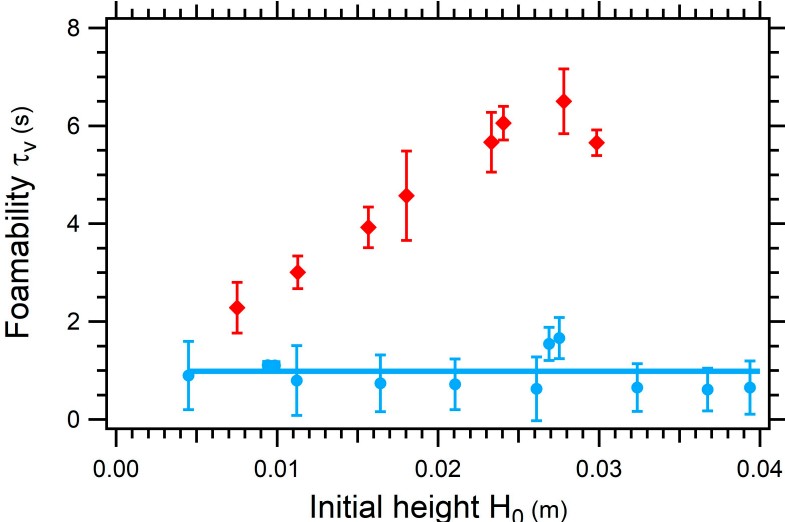

**Figure 12.** Foamability $\tau_v$ as a function of the initial height $H_0$ of liquid introduced in the glass column: decane/toluene (45/55 wt%) (blue circles) and decane/toluene (45/55 wt%) with 5 wt% of asphaltenes TR (red diamonds). Solid lines are guide for the eye. The flow rate of nitrogen is equal to $1.04 \times 10^{-5}$ m$^3$/s. The full line corresponds to the value of the foamability of a decane/toluene solution obtained by averaging all the measured values.

A possible explanation is that a small quantity of asphaltenes is captured at the interface of the bubbles that rise through the liquid and increases the lifetime of the bubble. In that picture, the larger is the initial height $H_0$, the more asphaltenes can be captured at the surface of bubbles, thus increasing the stability of the foam, similarly to froth flotation of minerals [26]. The contribution of these molecules to the interfacial tension should nevertheless be very small. The decrease of interfacial tension for dilute objects being RT$\Gamma$, where RT is the thermal energy and $\Gamma$ the surface coverage in mole per square meter. A coverage of one aggregate per 1000 nm$^2$ would lead to a decrease of the surface tension of $4 \times 10^{-3}$ mN/m, so clearly not significant in terms of surface tension and close to the lower detection limit of the Marangoni method.

Therefore, we suggest that a small quantity of asphaltenes are trapped at the interface, that they do not contribute to the interfacial energy, but that during the drainage of the liquid films within the foam, they are trapped in the liquid films and then behave as a barrier against coalescence of bubbles (Figure 13).

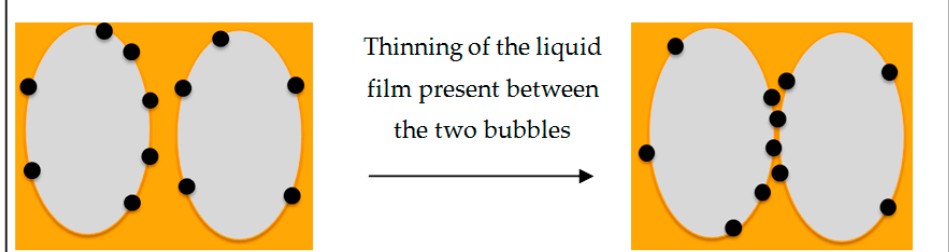

**Figure 13.** Schematical reprensentation of the foam stabilitsation mechanism induced by the presence of asphaltenes.

## 4. Conclusions

In this paper, we have shown that the investigated asphaltenes do not induce significant modification of the surface tension of decane/toluene mixtures at short time scales (<100 s) for asphaltenes concentrations up to 5 wt%. Both pending drop tensiometry and a Marangoni flow method were used, the latter technique allowing the detection of very small relative surface tension variations, down to a few $10^{-2}$%. That result is in agreement with previous reports in the literature. At longer times, we observe in the pending drop experiment the formation of a solid layer, which is attributed in the literature to a slow adsorption process of the asphaltenes at oil/air interfaces. However, we demonstrate that it rather results from the evaporation of the solvent and can be inhibited by saturating the atmosphere. In summary, we conclude that the investigated asphaltenes have no significant surface active properties, either at short or long times.

Nevertheless, the asphaltenes do increase the foamability of decane/toluene mixtures, that effect being significant for concentrations above 1 wt%. The increase is maximum at the precipitation limit of the asphaltenes and the same behavior was observed for two asphaltenes of different natures. In addition, we show that the non-zero foamability of the oil mixture has to be accounted for in order to properly quantify the effect of asphaltenes on foamability. That point had never been addressed in the literature, and could explain part of the observed foamability increases that have been attributed to the presence of asphaltenes only. In the case of oil mixtures containing asphaltenes, we show that the foamability surprisingly depends on the initial height of the liquid. A mechanism of stabilization of interfaces by asphaltenes is suggested, in which the bubbles rising in the liquid capture at their interface the asphaltenes present in solution. During the drainage of the liquid films within the foams, the asphaltenes trapped in the liquid films could then behave as a barrier against coalescence of bubbles, even if the contribution to surface energy is negligible. The hypothesis that asphaltenes may adsorb at air interface but do not lower the surface tension, is consistent with the large size of the aggregates they can form, leading to smaller surface pressures than the ones resulting from the adsorption of large quantities of smaller species such as surfactants.

**Author Contributions:** Investigation, M.A.; Supervision, J.-P.G., N.P.-B., F.L. and L.T.

**Funding:** This research was funded by Total S.A.

**Conflicts of Interest:** We declare no conflicts of interest.

## Appendix A

In the following appendix we evaluate the foam height $H_m$ and the lifetime $\tau_v$ of a bubble arriving at the top of the glass column during a foamability test.

The foam height $H_m$ is equal to the following expression, $H_f$ being the final height and $H_l$ the height of the bubbly liquid:

$$H_m = H_f - H_l \tag{A1}$$

If we consider a column of foam of radius $R$, a liquid volume fraction of foam in the upper part equal to $\varphi_l$ and a volume fraction of gas in the bubbly liquid $A$, the volume of liquid introduced initially $H_0 \pi R^2$ is distributed between:

- The upper part composed of foam which volume is equal to $H_m \varphi_l \, \pi R^2$
- The lower part composed of bubbly liquid which liquid volume is equal to $H_l(1 - A)\pi R^2$ Thus, we obtain:

$$H_0 = H_m \varphi_l + H_l(1 - A) \tag{A2}$$

$$\text{as } H_l = H_f - H_m$$

$$\text{then : } H_m = \frac{H_0 - H_f(1 - A)}{(\varphi_l + A - 1)} \tag{A3}$$

**Appendix B**

The lifetime $\tau_v$ of a bubble arriving at the top of the glass columns during a foamability test is defined by the following equation

$$\tau_v = \frac{H_m}{U} = \frac{\pi R^2}{Q}\left(\frac{H_f(1 - A) - H_0}{1 - A - \varphi_l}\right) \tag{A4}$$

With $U$ the speed of the flow in the column, $R$ the radius of the glass column, $Q$ the flow rate of nitrogen injection, $A$ the volume fraction of gas in the bubbly liquid and $\varphi_l$ the volume fraction of liquid in the foam.

The calculation of the lifetime $\tau_v$ requires to know the value of the foam height $H_m$ that-is-to-say the expression of the gas volume fraction $A$ in the bubbly liquid. This magnitude can be estimated using foamability measurements on a non-foaming liquid. Indeed, in the case of a non-foaming liquid, the foam height $H_m$ is equal to zero and the injection of nitrogen induces only an increase of the liquid height.

$$\tau_v = 0 \leftrightarrow H_m = 0$$

Thus, we obtained the following expression using the Equation (A3):

$$H_f = \frac{H_0}{1 - A} \tag{A5}$$

Which is equal to:

$$A = \frac{Q}{v_{lim}\pi R^2} = \frac{H_f - H_0}{H_0} \tag{A6}$$

We can therefore calibrate the magnitude $A$ by measuring the initial height $H_0$ and the final height $H_f$ for a system that does not foam, for instance toluene.

The measurements of the gas volume fraction $A$ as a function of the flow rate $Q$ at a fixed initial height $H_0$ is represented in Figure A1. The measurements of the gas volume fraction $A$ as a function of initial height $H_0$ at a fixed flow rate $Q$ is represented in Figure A2.

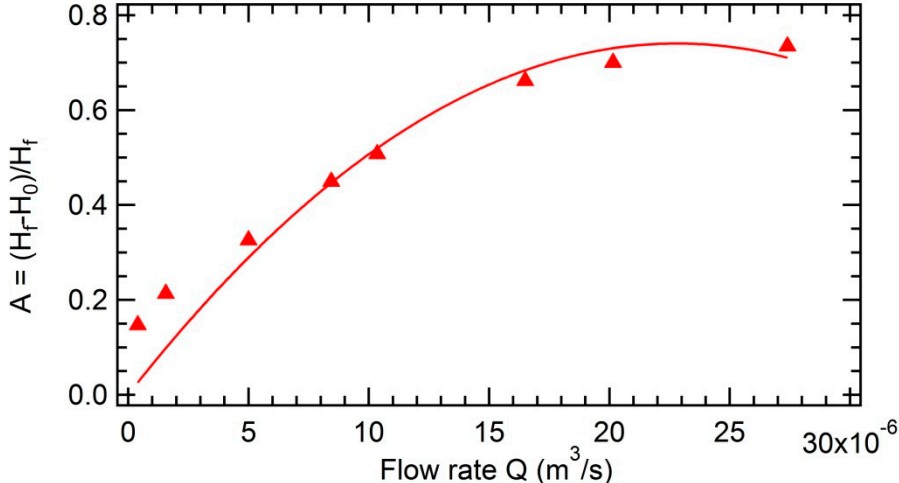

**Figure A1.** Evolution of the gas volume fraction A in the bubbly liquid as a function of the flow rate $Q$ for a toluene sample. $H_0 = 1.18$ cm.

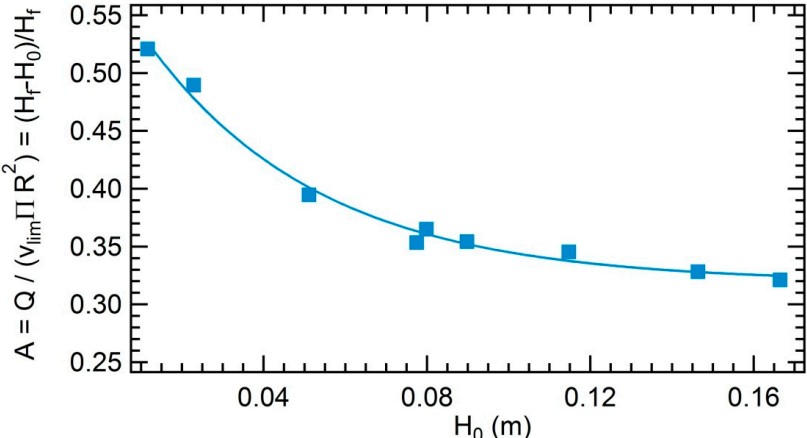

**Figure A2.** Evolution of the gas volume fraction A in the bubbly liquid as a function of the the initial height $H_0$ of liquid introduced in the glass column. $Q = 1.04 \times 10^{-5} m^3/s$

Experimentally, we obtain the following empirical laws:

$$A_{|H_0}(Q) = 6.5 \times 10^4 Q - 1.4 \times 10^9 Q^2 \tag{A7}$$

with $H_0 = 1.18$ cm

$$A_{|Q}(H_0) = 3.2 \times 10^{-1} + 2.7 \times 10^{-1} \exp(-2.4 \times 10^1 H_0) \tag{A8}$$

with $Q = 1.04 \times 10^{-5} m^3/s$

We should keep in mind that in the case of another system that does not foam, the value of $A(Q, H_0)$ evaluated previously can sometimes induce lifetimes $\tau_v$ slightly negative. We are, indeed, in an area where the noise of the measurements prevails. By convention, we will consider these "negative" lifetimes as equal to zero.

## Appendix C

The expression of the liquid volume fraction $\varphi_l$ is obtained from the general equation of drainage, which expression, by neglecting the capillary suction, is:

$$\frac{D\varphi_l}{Dt} + \vec{\nabla}\left(\frac{\alpha\rho_l}{\eta}\vec{g}\right) = 0 \tag{A9}$$

with $\alpha$ the permeability coefficient of the foam.

By projecting the previous equation along the vertical axis $z$ oriented upwards, we obtain the following expression in steady state regime:

$$U\frac{d\varphi_l}{dz} - \frac{\partial}{\partial z}\left(\frac{\alpha\rho_l g}{\eta}\right) = 0 \tag{A10}$$

that-is-to-say:

$$\frac{\partial}{\partial z}\left(U\varphi_l - \frac{\alpha\rho_l g}{\eta}\right) = 0 \tag{A11}$$

The term in parenthesis corresponds to the flux, which is equal to zero at the interface. Thus, we obtain the following relation:

$$U\varphi_l = \frac{\alpha\rho_l g}{\eta} \tag{A12}$$

That-is-to-say:

$$\varphi_l = \frac{\alpha\rho_l g}{\eta U} \tag{A13}$$

Please note that there are several models in the literature to describe the coefficient of permeability $\alpha$. We choose the model of Carman and Kozeny which is an empirical equation for the permeability of an assembly of spheres providing an expression of the coefficient of permeability $\alpha$ in reasonable adequacy with the experiments ($\varphi_l > 0.1$):

$$\alpha = \frac{\varphi_l{}^3 d^2}{180(1-\varphi_l)^2} \tag{A14}$$

We recall that $d$ is the diameter of a bubble: typically $d = 3.2$ mm in toluene at a nitrogen injection rate of $1.04 \times 10^{-5}$ m$^3$/s.

Thus,

$$\varphi_l = \frac{\alpha\rho_l g}{\eta} = \frac{\varphi_l{}^3 d^2}{180(1-\varphi_l)^2} \cdot \frac{\rho_l g}{\eta} \tag{A15}$$

By replacing the expression of $\alpha$ in the previous equation, we obtain the expression of the liquid volume fraction $\varphi_l$:

$$\varphi_l = \left(\frac{180U}{d^2\frac{\rho_l g}{\eta} + 180U}\right)^{\frac{1}{2}} \tag{A16}$$

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
