# Peer review of "Asphaltenes at Oil/Gas Interfaces: Foamability Even with No Significant Surface Activity"

_colloids, doi:10.3390/colloids3010002_

Round 1

Reviewer 1 Report

No further comments!

Author Response

We thank the reviewer for his/her careful reading of the manuscript.

Reviewer 2 Report

The revised manuscript presents significant improvements in writing style, science, and the analysis of results. I have a few suggestions that could enhance the revised version. 

Introduction: the authors state: "Crude oils are complex mixtures of hydrocarbons and various organic compounds [1–3] including asphaltenes, which are polyaromatic molecules with side alkane chains." I recommend acknowledging the controversy on asphaltene structure. The authors could say something like "asphaltenes are polyaromatic species which structure has remained controversial during several decades. There are two proposed structural models: the classical island model, where asphaltenes exhibit only one aromatic core with alkyl side chains, and the more recently archipelago model, where asphaltenes exhibit several aromatic cores interconnected by aryl linkages, naphthenic bridges, and alkyl-side chains. Asphaltene behavior suggests that both structures, island and archipelago, coexist." Please, include references by Oliver C. Mullins regarding the island model. Recent and well-cited work by Chacon-Patino deals with the archipelago model and demonstrate the reason behind contradictory results (Energy Fuels 2017, 311213509-13518). Asphaltene properties, including interfacial behavior, are the result of the molecular structure. Therefore, it is crucial that the authors discuss structure in the introduction. 

Asphaltenes from different crude oils exhibit different properties. It is essential that the authors discuss in detail bulk properties of the crude oils and the asphaltene fractions. For instance, the percentage of saturates/aromatics/resins/asphaltenes, bulk elemental composition with focus on O/C and H/C ratios for asphaltenes. 

In general, the authors improved the quality of the manuscript. I strongly recommend the publication of the corrected version, since the work is novel, relevant to the field, and demonstrates that different asphaltene samples exhibit different properties. This report supports the hypothesis that a solely generalized structural model (island) cannot be used to describe all asphaltene samples. Structure governs function, and this work clearly demonstrates that geological origin affects interfacial behavior. I recommend that the authors add a few comments on the structural/function discussion. 

Author Response

We thank the reviewers for their careful reading of the manuscript. We have modified the manuscript according to reviewer 2’s comments and reply below.

In addition, the manuscript has been read by an English speaking colleague for further editing improvement.

The revised manuscript presents significant improvements in writing style, science, and the analysis of results. I have a few suggestions that could enhance the revised version. 

Introduction: the authors state: "Crude oils are complex mixtures of hydrocarbons and various organic compounds [1–3] including asphaltenes, which are polyaromatic molecules with side alkane chains." I recommend acknowledging the controversy on asphaltene structure. The authors could say something like "asphaltenes are polyaromatic species which structure has remained controversial during several decades. There are two proposed structural models: the classical island model, where asphaltenes exhibit only one aromatic core with alkyl side chains, and the more recently archipelago model, where asphaltenes exhibit several aromatic cores interconnected by aryl linkages, naphthenic bridges, and alkyl-side chains. Asphaltene behavior suggests that both structures, island and archipelago, coexist." Please, include references by Oliver C. Mullins regarding the island model. Recent and well-cited work by Chacon-Patino deals with the archipelago model and demonstrate the reason behind contradictory results (Energy Fuels 2017, 31, 12, 13509-13518). Asphaltene properties, including interfacial behavior, are the result of the molecular structure. Therefore, it is crucial that the authors discuss structure in the introduction. 

Asphaltenes from different crude oils exhibit different properties. It is essential that the authors discuss in detail bulk properties of the crude oils and the asphaltene fractions. For instance, the percentage of saturates/aromatics/resins/asphaltenes, bulk elemental composition with focus on O/C and H/C ratios for asphaltenes. 

In general, the authors improved the quality of the manuscript. I strongly recommend the publication of the corrected version, since the work is novel, relevant to the field, and demonstrates that different asphaltene samples exhibit different properties. This report supports the hypothesis that a solely generalized structural model (island) cannot be used to describe all asphaltene samples. Structure governs function, and this work clearly demonstrates that geological origin affects interfacial behavior. I recommend that the authors add a few comments on the structural/function discussion. 

We thank the reviewer for his very relevant comments on the debate on the structure models. We have modified the introduction and added references to the works by Chacon-Patino and Mullins as suggested by the reviewer (references 5 and 6). In addition, we now make reference to previously published data on the asphaltenes we use (line 121). However, we think more details on that point would be out of the scope of our work, since in particular there is no clearly established link between the structural models and interfacial properties.

This manuscript is a resubmission of an earlier submission. The following is a list of the peer review reports and author responses from that submission.

Round 1

Reviewer 1 Report

Here the authors studied asphaltenes at oil/gas interface.  Although they stated that they were mainly interested in studying the mechanisms of asphaltenes at the interface, I found little discussion on the mechanisms. I also wonder on how much the systems selected here are to real crude oil. Thus I recommend further work here to make the paper more scientific.

1.       Title might be tweaked a little.  Maybe Foamability even with no significant surface activity

2.       The first sentence of the abstract doesn’t make sense: I don’t see crude oil in gas stations

3.       I recommend more phase behavior experiments of asphaltenes in solvents to determine precipitation rates…etc

4.       Page 5: I don’t see any data for surface tension over time. It is not enough to say it didn’t change

5.       Figure 3. It is not obvious which drop is at 2500s.

6.       Figure 9 and 10, the x axis should be weight percentage of decane…etc

7.       Figure 11. Was the x axis units in meters?

8.       I recommend discussing mechanisms better int his paper. Please refer to other papers in the literature that discuss bulk oil foam in details. Ex. https://doi.org/10.1016/j.jcis.2018.04.056

Reviewer 2 Report

This paper report an interesting research about effect of asphaltene in several surface properties of a synthetic mixture. Introduction is quite clarifying and summarize the previous description of the topics to be covered. Involved experimental techniques are several and appears adequate for subject of study. Obtained results are relevant, confirm previous known facts and offer some interesting and new descriptions.

The purpose of the paper fit well with the journal Colloids and Interfaces and Consequently my recommendation is to ACCEPT the paper for publication.

However I would list some points that should be considered as minor revision in order to clarify the described work:

-      The experimental conditions (solvents ratio, asphaltene percentage) covered with the several techniques are not the same. Even when it is indicated along the text, in order to clarify, it could be interesting a table with the summary of such conditions and the involved technique.

-      Demonstration of skin formation by evaporation of solvent appears quite clear. It could be interesting to have some experiment where toluene is replaced by a heavy compound (for instance naphthalene), if authors carried out some experimentation in such direction it could be reported.

-      There is a mistake in the numeration and 3.3 has to be changed to 3.2.

-      According to my own experience, reproducibility of the experiment of foamability is usually poor. Figs 9 and 10 require additional information about error bars in order to quantify if presence of asphaltene is relevant or not. Please note how author state that foamability for asphaltene-free samples is “unexpectedly non zero”. Such non zero value is as important as 50% (related to mixtures with asphaltene) for some concentrations.

-      It should be explained why Figure 11 does not cover the same H0 range for systems with and without asphaltene and to clarify if the shown difference is occurring only at a short H0 range.

-      H0 effect could be related with the system of N2 injection and its depth. Please clarify if such point was experimentally checked.

Reviewer 3 Report

This paper reports on “Asphaltenes at oil/gas interfaces: Foamability with no significant surface activity”. After carefully looked through the manuscript, it might be suitable to be accepted if authors cover below points in order to have a clearer manuscript:

1-     There are some similar studies reported in the literature. Some of them have been highlighted in the introduction section. So the originality of the paper needs to be further clarified.

2-     Editing and proofreading are required throughout the manuscript.

3-     Most of references must be updated within the past 5 years.

4-     Lines, 87-97 authors summarised some literatures that asphaltenes clusters behave like proteins. Can authors give a possible chemical and physical mechanism responsible for foamability of asphaltenes at oil/gas for their study?

5-     Can authors make a comparison between the finding in this study and the outcomes from the study reported in Journal of Colloid and Interface Science 2001, 239, 501–508?

6-     Line 168 please provide some dynamic images for supporting your results.

7-     Line 236-237, please add references.

8-     Line 260, what the black star in Figure 6 stand for?

9-     Error bars needed on figures 6-11.

10- There is an inappreciate focus on the comparison results of this work with the published work.

Reviewer 4 Report

The authors make over-conclusions about asphaltene surface properties. From this reviewer point of view, the manuscript needs to be re-written. Also, the authors need to reference the work the extensive work on asphaltene interfacial properties by H. Yarranton. Below there are some specific comments:

Page 1, line 18: recommend writing “enriched with asphaltenes” instead of “containing large amount of asphaltenes”. 

Page 1, line 19: In this research/study/work, instead of “paper”.

Page 1, lines 21-22: there is an apparent contradiction to all the work reported, especially by Harvey Yarranton et al. Asphaltenes have been demonstrated to present surface activity, and the degree of these properties is geological dependent. The authors need to correct this statement; otherwise, the manuscript cannot be published. How many crude oils the authors analyzed? General conclusions cannot be drawn on a few samples. For instance, Athabasca Bitumen asphaltenes are well recognized as an excellent example for asphaltene surface-active properties. Indeed, it is likely that surface activity arises from heteroatomic and polarizable functionalities. 

Page 2, line 53: Please, I recommend including the following references: Energy Fuels 2015, 29, 3, 1323-1331, Energy Fuels 2014, 28, 5, 2831-2856, Energy Fuels 2018, 32, 7, 7347-7357.

Page 5, lines 192-200: the skin observed by the authors, or I will refer has “external layer” is due perhaps to asphaltene aggregation at the interface? This phenomenon has been previously documented Energy Fuels 2007, 21, 4, 2129-2137.

Page 6, lines 208-211. The effect of solvent evaporation is minimized by the continuous injection of solvent to keep the volume constant. It is likely that under these conditions, asphaltene aggregation and behavior at the interface play a significant role. I recommend re-writing the whole argument. 

Figure 5. I recommend replacing “asphaltene skin” by “asphaltene layer”. 
Page 7, line 253. Please, rewrite as: “The foamability will be defined later.”

Page 8, lines 272-273. The authors state: “This confirms that asphaltenes do not have any surface active property at the oil/air interface…” It is short-sighted to make “universal” statements, especially using words as “any”, “all”, etc. From my point of view, the specific asphaltene samples studied in this report exhibit decreased surface active properties at the oil/air interface, when compared with asphaltenes from other geological sources, especially at the oil/water interface. The authors need to re-write this kind of statements abundant throughout the manuscript.